# Condylar Changes in Children with Posterior Crossbite after Maxillary Expansion: Tridimensional Evaluation

**DOI:** 10.3390/children8010038

**Published:** 2021-01-11

**Authors:** Rosamaria Fastuca, Helga Turiaco, Fausto Assandri, Piero A. Zecca, Luca Levrini, Alberto Caprioglio

**Affiliations:** 1Department of Biomedical, Surgical and Dental Sciences, University of Milan, 20122 Milan, Italy; alberto.caprioglio@unimi.it; 2Department of Orthodontics, University of Insubria, 21100 Varese, Italy; helga_turiaco@virgilio.it; 3MD-MS, UOC Maxillo-Facial Surgery and Dentistry, Fondazione IRCCS Cà Granda, Ospedale Maggiore Policlinico, 20122 Milan, Italy; fausto.assandri@unimi.it; 4Department of Medicine and Surgery, University of Insubria, 21100 Varese, Italy; pieroantonio@gmail.com (P.A.Z.); luca.levrini@uninsubria.it (L.L.)

**Keywords:** crossbite, maxillary expansion, cone beam computed tomography

## Abstract

(1) Background: To investigate condylar position in subjects with functional posterior crossbite comparing findings before and after rapid maxillary expansion (RME) treatment through 3D analysis; (2) Methods: Thirty-two Caucasian patients (14 males, mean age 8 y 8 m ± 1 y 2 m; 18 females mean age 8 y 2 m ± 1 y 4 m) with functional posterior crossbite (FPXB) diagnosis underwent rapid palatal expansion with a Haas appliance banded on second deciduous upper molars. Patients’ underwent CBCT scans before rapid palatal expansion (T0) and after 12 months (T1). The images were processed through 3D slicer software; (3) Results: The condylar position changes between T1 and T0 among the crossbite and non-crossbite sides were not statistically significant, except for the transversal axis. At T1, the condyles moved forward (y axis) and laterally (x axis), they also moved downward (z axis) but not significantly; (4) Conclusions: Condilar position in growing patients with functional posterior crossbite did not change significantly after rapid maxillary expansion.

## 1. Introduction

Posterior crossbite is one of the most common malocclusion orthodontists have to deal with during their daily practice, with a reported prevalence from 7% to 23% [1,2]. The most common type of posterior crossbite is a unilateral one. [3] The functional posterior crossbite (FPXB) represents 80% to 97% of all the posterior crossbites [4,5] and is characterized by a discrepancy between the centric occlusion (CO) and the centric relation (CR). Patients with FPXB might present an asymmetrical position of the mandibular condyles that, during growth, if left uncorrected, might lead to morphological asymmetry [3,4,5,6].

One of the most common appliances used to early correct this type of malocclusion is rapid maxillary expansion (RME) [7], since an FPXB in growing children might be related to premature contacts on deciduous teeth due to maxillary transverse deficiency leading to mandibular shift in order to accommodate occlusion. After maxillary expansion, transverse, vertical and anteroposterior changes of maxilla and mandible were reported, such as clockwise rotation of the mandibular plane, resulting from a lower and posterior position of the mandible in unilateral crossbites and sagittal changes in bilateral crossbites [7,8]. The protocol foresees the possibility to apply the maxillary expander both on the first permanent molars or the second deciduous molars, but the maxillary expansion on the deciduous molars showed some advantages, such as a spontaneous rotation of the first molars and a spontaneous retraction and alignment of the upper central and lateral incisors compared to the expansion on the first molars [9] and positive effects of treatment on canine inclination such as a reduction of α-angle [10]. Moreover, the expansion of permanent molars could affect their periodontal ligament and endodontic status [11,12].

As already demonstrated by several authors [13,14], maxillary expansion is based on the separation of the midpalatal suture with a following transversal expansion; however, the high forces expressed during the process also caused changes in the surrounding frontomaxillary, zygomaticomaxillary, zygomaticotemporal and pterygopalatine sutures. Moreover, while the effects of RME are well-documented, little is known about its effects on the condyles, which are not in the peak growth phase yet between 6 and 9 years of age [15]. Other studies have already reported how the condyles and the glenoid-fossa can undergo bone remodeling after some orthopedic interventions causing changes in the condilar position [16,17].

Several studies suggested that patients with FPXB also presented asymmetric condyle position and mandibular displacements that were solved after RME treatment [1,18,19]. On the other hand, other studies reported a symmetrical condyle position before and after treatment with RME [20,21,22]. Most of the studies performed over the past years were based on 2D analysis [1,2,3,4,5,6,7,8,9,10,11,12,13,14,15,16,17,18,19,20,21,22,23] or magnetic resonance [24,25], while 3D analysis is of more recent use [20,21,22], which could partly be the reason of the discordance between the results of the different studies. 

The aim of the present study was to investigate if patients with FPXB present an asymmetrical position of the mandibular condyles before treatment and evaluate changes in their position after RME treatment. The null hypothesis was that RME does not induce immediate changes in the position of the condyles in Class I malocclusion patients with FPXB.

## 2. Materials and Methods

The initial sample of the present retrospective study consisted of patients treated with RME, selected from the Departments of Orthodontics of University of Insubria ‘Ospedale del Circolo Fondazione Macchi’ (Varese, Italy) and in a private practice of a Board Certified Specialist. Signed informed consent for releasing diagnostic records for scientific purposes was available from the parents of the patients. Protocol was reviewed and approved by the Ethical Committee (Approval no. 826) and procedures followed adhered to the World Medical Organization Declaration of Helsinki. 

The inclusion criteria were as follows: good general health, early mixed dentition, stage 1 or 2 of cervical vertebral maturation (CVM), transverse maxillary deficiency with FPXB treated by using RME, skeletal Class I, age between 6 and 9 years old and availability of complete initial and final records, including CBCT scans, photographs, dental casts and medical history forms.

Exclusion criteria comprised systemic diseases and craniofacial syndromes, severe facial asymmetry, dental anomalies, stage 3 or more of CVM and a history of other orthodontic treatment prior to RME. From the initial sample of 60 patients, 32 patients (14 males, mean age 8 y 8 m ± 1 y 2 m; 18 females mean age 8 y 2 m ± 1 y 4 m), treated between January 2015 and September 2017, were included in the study.

All patients underwent RME with a Haas-type expander (Snap Lock Expander 10 mm A167-1439, Forestadent, Pforzheim, Germany). The maxillary expanders were all banded on the second deciduous molars using a glass ionomer cement in accordance with the manufacturer’s instructions. After the maxillary expander was correctly positioned, two activations (0.45 mm) were performed by the dental practitioner, whereas in the following days, a single activation (0.225 mm activation per day) was performed until lingual cusps of the upper first molars occluded onto the lingual side of the buccal cuspids of the lower first molars. The screw was then locked with light-cure flow composite, and the expander was kept on the teeth as a passive retainer. The overall treatment time lasted 10.2  ±  2 months and comprised active treatment and retention.

CBCT scans (i-CAT Classic, Imaging Sc. Int., Hatfield, PA, USA) were taken at the beginning of the treatment, before the maxillary expander was put in place (T0), and at the end of the treatment (T1), after the appliance was removed. 

### 2.1. Image Analysis

The dicom images were manipulated into Slicer software (https://www.slicer.org/; release 4.10.2) [26] in order to perform the identification of a reliable set of landmarks. For every temporomandibular joint (TMJ), the following landmarks were defined and placed: Co: condylion; Post: outlet of Civinini’s (Huguier’s) canal (canaliculus in the Glaser fissure allowing the exit of the chorda tympani from the tympanic cavity); Emi: most anterior and inferior point of the articular eminence (Figure 1). Additionally, the landmarks to identify a reference plane were identified: the most inferior point of the sella turcica; the most distal point of the border of the left and right foramen ovale. Then, using the Rhinoceros software (McNeel, R., and others. (2010). Rhinoceros 3D, Version 6.0. Robert McNeel & Associates, Seattle, WA, USA), changes in the position of the landmarks on the x, y and z axes were evaluated, and the distances between the landmarks on the left and right side were measured.

### 2.2. Statistical Analysis 

Sample size was calculated on the measurements of three patients, selecting as the main outcome the maxillary skeletal changes before and after treatment (PaFR-PaFL, Palatal foramen right-Palatal foramen left distance). A sample size of at least 15 subjects was necessary to detect a power of 0.8.

The SPSS software, version 23.0 (SPSS^®^ Inc., Chicago, IL, USA) was used to perform the statistical analyses. The Shapiro-Wilk test and Levene test confirmed the normal distributions and equal variances between T0 and T1, respectively. Means and standard deviations (SD) were computed for all the variables.

A paired *t*-test was employed to compare the landmarks position on both sides (XB and non-XB) between the timepoints. A Student’s *t*-test was employed to compare changes in the landmarks’ position between Xb and non-XB side. A *p*-value less than 0.05 was used in the rejection of the null hypothesis.

### 2.3. Method Error

The same trained operator (PZ) performed and repeated the measurements two weeks later for five patients. Systematic and random errors were calculated comparing the first and second measurements with paired *t*-tests and Dahlberg’s formula, at a significance level of *p* < 0.05 [27]. All measurement error coefficients were found to be adequate for appropriate reproducibility of the study.

## 3. Results

In all the cases, the efficacy of the separation of the midpalatal suture was demonstrated, assessed as the PaFR-PaFL distance, showed an increase of 2.5 ± 0.2 mm (mean and SD).

A total of 64 CBCTs were studied in the patient group (32 at T0 and 32 at T1): 19 patients had a right FPXB and 13 patients had a left FPXB.

Means and SDs for the two time points and results of the paired *t*-test are shown for the position of the condyles and the changes in the articular fossa in Table 1. 

The outcomes at T1 showed significant changes in Condylion position along the x and y axes for both sides. Moreover, distance between the condyles and the articular fossae significantly increased after treatment.

Means and SDs for the changes comparing XB and Non-XB side are reported in Table 2 and Figure 2.

Only the Condylion position along the x axis showed a significant difference between sides.

## 4. Discussion

Other studies have already analyzed the changes of the condylar position after the expansion of the first permanent molars, but none has been performed on the expansion of the primary second molars. 

Matta et al. [18] analyzed the condyle position before treatment and after 12 weeks with conventional tomography and, according to their study, the condyles presented an asymmetrical position that was altered by RME, promoting greater spatial symmetry between the two sides.

Leonardi et al. [19] analyzed condyle position with conventional tomography after expansion on first permanent molars in FPXB patients and their results showed that the condyles were symmetrical before RME and the treatment promoted an increase of the joint spaces maintaining the symmetry of the condyle-fossa joint relationship.

Melgaço et al. [20] and Ghoussub et al. [21] assessed the condylar response before and after expansion of the permanent molars using CBCT scans in patients in need of expansion, but without a crossbite, and in patients with bilateral crossbite, respectively. In both cases, the condylar position was symmetrical before and after treatment, but it underwent some positional changes.

McLeod et al. [28] evaluated condyles before and after maxillary expansion of the first permanent molars in FPXB patients, and no statistically significant changes were found, so RME did not alter the condylar position.

Previous studies [29,30] have already reported that mandibular repositioning can be expected after RME in Class II patients, but only Class I patients were taken into account for this study, which could have influenced the final results.

According to our results, the condylar position before treatment in growing patients with FPXB is not symmetrical, and this condition is improved after treatment.

In this study, the condyles at T1 (after 1 year) and the articular fossa presented some significant changes: analyzing the data showed that the condyles moved forward (y axis) and expanded (x axis); they also moved downward (z axis), but not significantly. In the articular fossa, an increase of the antero-posterior dimension was noticed. However, if we compare the changes that occurred between T1 and T0 among the two sides (XB and non-XB), no statistically significant different movement occurred, except for the displacements that took place along the transversal axis. According to our data, a major displacement on the non-XB side occurred along the transversal axis, which could be caused by the association of the growth of the patient with a condylar repositioning toward the non-XB side. 

Even if the results are statistically significant, the changes are slight and clinically not appreciable. Nevertheless, minor changes at the condylar level might reflect greater changes in occlusion, allowing a transversal shift of the mandible, which might change its position, leading to more symmetrical occlusion after RME if compared to the pre-treatment position. These changes often occur during RME treatment and are clinically desired in order to improve the midlines, solve the crossbite and allow better occlusal contacts. 

The discrepancy between the results with some previous studies could be partly justified because of the different methods used, such as the magnetic resonance used by Arat et al. [23,24] or the transcranial radiographs used by Kecik et al. [1] and Myers et al. [22]. On the other hand, this study takes into account only positional measurements and not the growth of the patient, as some other studies have [20,21,22,23,24,25,26,27,28]; for this reason our results cannot be compared with theirs.

The lack of a control group, for ethical reasons, may limit the outcomes of the study. Furthermore, the lack of a control group does not allow to verify if the modifications recorded may be occurred by remodeling of the condyles during the growth of the patient. The present study was performed in a short-term period (1 year), and the short interval time might be biased by the results, emphasizing the immediate changes around maxillary structures, which are directly influenced by maxillary expansion. Long-term studies are further needed in order to investigate if condyle remodeling does not occur and to ensure the stability of the results.

## 5. Conclusions

Based on the results of this study, it can be assessed that:In growing patients with unilateral posterior functional crossbite, changes in condylar position after RME between crossbite and non-crossbite sides were not clinically significant.RME treatment is recommended when maxillary deficiency is present but is not always associated with a clinically significant reposition of the mandibular condyles, even though small changes at the condyles might reflect in changes in occlusion, which were not considered in the present study.

## Figures and Tables

**Figure 1 children-08-00038-f001:**
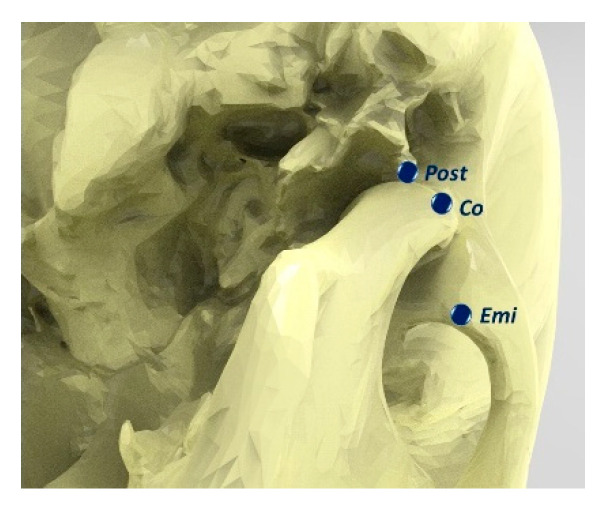
3D landmarks placement: Co: condylion; Post: outlet of the Civinini’s (Huguier’s) canal (canaliculus in the Glaser fissure allowing the exit of the chorda tympani from the tympanic cavity); Emi: most anterior and inferior point of the articular eminence.

**Figure 2 children-08-00038-f002:**
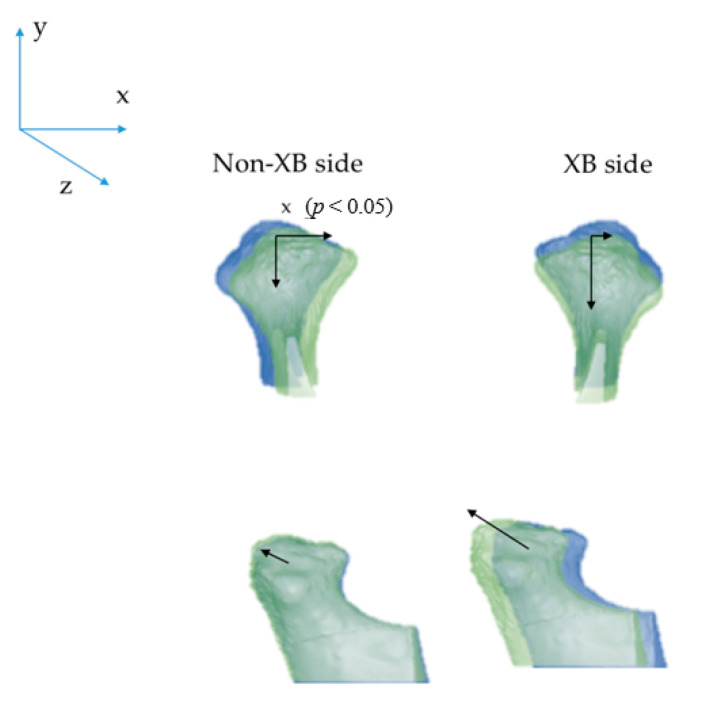
Condylar changes (T1–T0). T0: blue; T1: green. Significant changes were found only for the *x*-axis between the Non-XB side and XB side.

**Table 1 children-08-00038-t001:** Condylar and articular fossa displacements at T0 and T1.

Variable	T0	T1	Paired *t*-Test
CoXB-x (mm)	46.24 ± 1.89	47.02 ± 2.18	0.01 *
CoXB-y (mm)	10.04 ± 3.10	13.42 ± 3.79	0.00 *
CoXB-z (mm)	−19.93 ± 3.08	−21.39 ± 3.60	0.06
Co-x (mm)	43.98 ± 2.06	46.06 ± 2.15	0.00 *
Co-y (mm)	10.93 ± 3.84	13.62 ± 2.66	0.00 *
Co-z (mm)	−20.72 ± 3.05	−20.88 ± 3.16	0.75
CoXB-Co (mm)	90.56 ± 3.81	92.28 ± 4.21	0.00 *
CoXB-EmiXB (mm)	10.17 ± 1.99	9.36 ± 1.73	0.09
Co-Emi (mm)	9.94 ± 1.73	8.99 ± 1.57	0.01 *
PostXB-EmiXB (mm)	16.36 ± 1.52	14.91 ± 2.01	0.00 *
Post-Emi (mm)	16.50 ± 2.97	15.43 ± 2.48	0.02 *
CoXB-PostXB (mm)	8.12 ± 1.36	8.38 ± 0.84	0.27
Co-Post (mm)	9.24 ± 2.71	9.38 ± 2.19	0.65
PostXB-Post (mm)	92.62 ± 5.58	96.55 ± 5.75	0.00 *
EmiXB-Emi (mm)	86.82 ± 4.68	90.37 ± 4.88	0.00 *

* *p* < 0.05.

**Table 2 children-08-00038-t002:** Condylar and articular fossa changes (T1–T0).

Variable	Non-XB		XB		Student *t*-Test
	Mean	SD	Mean	SD	
Co-x (mm)	2.08	1.34	0.78	0.97	0.01 *
Co-y (mm)	2.69	2.21	3.38	2.43	0.44
Co-z (mm)	−0.15	1.80	−1.46	2.61	0.14
Co-Emi (mm)	−0.95	1.19	−0.81	1.64	0.81
Post-Emi (mm)	−1.07	1.46	−1.45	1.42	0.49
Co-Post (mm)	0.14	1.11	0.26	0.86	0.74

Comparison of XB and non-XB side. Data are shown as mean ± SD. Levels of significance, *p* < 0.05 *.

## Data Availability

Data available on request due to restrictions e.g., privacy or ethical.

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
