# Peer review of "Condylar Changes in Children with Posterior Crossbite after Maxillary Expansion: Tridimensional Evaluation"

_children, 2021, doi:10.3390/children8010038_

Round 1
Reviewer 1 Report
Page 2, line 89 – The authors should remove the trade name of the cement. Page 2, line 9 -– The authors should remove the trade name of the composite In the discussion part, please add the (clinical) relevance of the statistically significant change in the transversal axis (probably it is relevant if it was mentioned in the abstract). Please add during the conclusion part, the clinical relevance for readers (orthodontists and general practitioners).Author Response
Page 2, line 89 – The authors should remove the trade name of the cement. Page 2, line 9 -– The authors should remove the trade name of the composite
R: The text was modified according to Reviewer’s suggestions.
In the discussion part, please add the (clinical) relevance of the statistically significant change in the transversal axis (probably it is relevant if it was mentioned in the abstract).
R: A paragraph was added to discuss the clinical relevance of the results obtained in the Discussion section (Page 6, lines 187-191).
Please add during the conclusion part, the clinical relevance for readers (orthodontists and general practitioners).
R: A paragraph was added to discuss the clinical relevance of the results obtained in the Conclusions section (Page 6, lines 206-211).
Reviewer 2 Report
First of all, congratulate the authors for the content of the article.
I would like to know how they estimated the total treatment time. Or if on the contrary it was totally random.
In terms of results, it would be convenient to make some type of graph that allows visually observing the results obtained.
As for the conclusions, they should be clearer and more concise. Its reformulation would be necessary.
Author Response
I would like to know how they estimated the total treatment time. Or if on the contrary it was totally random.
R: Overall treatment time lasted 10.2 ± 2 months and comprised active treatment and retention. The data were obtained from the clinical records of the selected patients. A sentence was modified in Materials and Methods section (Page 2, line 95) in order to explain the overall treatment time.
In terms of results, it would be convenient to make some type of graph that allows visually observing the results obtained.
R: Figure 2 was added to the revised text summarizing the results.
As for the conclusions, they should be clearer and more concise. Its reformulation would be necessary.
R: The Conclusions section was entirely revised to be clearer, more concise and underline clinical relevance.
Round 2
Reviewer 2 Report
The authors are congratulated for the modifications madeThe authors are congratulated for the modifications made